# Expansion of the Nodal-Adjoint Method for Simple and Efficient Computation of the 2D Tomographic Imaging Jacobian Matrix

**DOI:** 10.3390/s21030729

**Published:** 2021-01-22

**Authors:** Samar Hosseinzadegan, Andreas Fhager, Mikael Persson, Shireen Geimer, Paul Meaney

**Affiliations:** 1Electrical Engineering Department, Chalmers University of Technology, 41296 Gothenburg, Sweden; samar.hosseinzadegan@gmail.com (S.H.); andreas.fhager@chalmers.se (A.F.); mikael.persson@chalmers.se (M.P.); 2Thayer School of Engineering, Dartmouth College, Hanover, NH 03755, USA; shireen.geimer@dartmouth.edu

**Keywords:** nodal adjoint method, Jacobian matrix, discrete dipole approximation, computational efficiency, microwave tomography, breast imaging

## Abstract

This paper focuses on the construction of the Jacobian matrix required in tomographic reconstruction algorithms. In microwave tomography, computing the forward solutions during the iterative reconstruction process impacts the accuracy and computational efficiency. Towards this end, we have applied the discrete dipole approximation for the forward solutions with significant time savings. However, while we have discovered that the imaging problem configuration can dramatically impact the computation time required for the forward solver, it can be equally beneficial in constructing the Jacobian matrix calculated in iterative image reconstruction algorithms. Key to this implementation, we propose to use the same simulation grid for both the forward and imaging domain discretizations for the discrete dipole approximation solutions and report in detail the theoretical aspects for this localization. In this way, the computational cost of the nodal adjoint method decreases by several orders of magnitude. Our investigations show that this expansion is a significant enhancement compared to previous implementations and results in a rapid calculation of the Jacobian matrix with a high level of accuracy. The discrete dipole approximation and the newly efficient Jacobian matrices are effectively implemented to produce quantitative images of the simplified breast phantom from the microwave imaging system.

## 1. Introduction

Microwave imaging is an emerging technology that is beginning to see increased clinical use for a number of applications. These include breast cancer imaging [1,2,3,4], stroke diagnosis [5,6], bone imaging [7] and others. For these applications, there is usually a significant endogenous contrast that can be exploited for imaging. For breast cancer imaging, there is a substantial property contrast between benign and malignant tissue [8,9,10]; for stroke diagnosis, there is a large contrast between blood and normal brain tissue [11]; for bone, there is a significant correlation between the dielectric properties and bone density [7]. In addition, the dielectric properties can be exploited to recover functional information such as temperature monitoring during thermal therapy [12,13] due to their temperature dependence [14].

The most widely applied area for microwave imaging has been in the area of breast cancer imaging. The three most prominent microwave imaging techniques include radar, holography and tomography or inverse problems. Radar approaches can be either in a backscatter or transmission mode, where broadband signals are transmitted and received by either the same antenna or others in an array. The signals are time shifted so as to focus the composite signals at a particular pixel in the field of view and the sum of these components is recorded. This procedure is repeated for all points in the domain until an image is produced. The work by Fear and Stuchly [4] is a good example of the reflection-based technique which has been used in simulation, phantom and limited patient exams. The Craddock group at the University of Bristol has produced good results with their transmission-based system which has been used in clinical trials [1,15]. Microwave holography generally assumes that the broadcast signal and those transmitted through and reflected back from the target are spherical waves which can be decomposed into a superposition of plane waves through a Fourier transform [16]. Each of these can be back propagated to the target and re-transformed into the spatial domain for target identification and characterization. These algorithms are effective in both 2D and 3D. The most prominent effort in this area is by the Nikolova group at McMaster University which has implemented a system that has advanced to phantom studies [17]. For the tomographic approach, numerous groups have produced simulation studies [18,19,20,21] on both classic conical structures and with data from actual MR exams [22]. Several approaches have advanced to phantom and animal studies [12,23,24,25,26] and there have been several patient studies by the Dartmouth College group—particularly in breast cancer neoadjuvant chemotherapy monitoring [2]. This paper focuses entirely on the tomography approaches since these are the ones that require construction of a Jacobian matrix as part of their image reconstruction process.

One of the consistent complaints about microwave tomography is the long computation times. This is due primarily to heavy resources needed for the multiple forward solution calculations at each iteration. These times have generally ranged from several minutes for some 2D implementations [27] to many hours and even days for larger 3D reconstructions that often require parallelized multi-processors and graphical processing units (GPUs) to accelerate the computation speed [18,23,24]. In fact, the EMTensor group is exploring massively parallel, cloud computing to solve the computational issues for their portable and computationally expensive system [28]. These concerns have been somewhat ignored because there are such significant challenges in just recovering a diagnostically relevant image. Problems have most often revolved around the issues of convergence to local minima [29], the need for a priori information by some systems [21,30], and the amount of measurement data necessary for reconstructing viable images and the associated hardware requirements to meet this need [31]. In most cases, it was generally assumed that the computational issues could be addressed later—especially since computer power has consistently improved by factors of two or so every 18 months for several decades now according to Moore’s Law [32].

While the forward solution computation costs are the most substantial, there are other aspects of these formulations which also contribute to the slow overall computation. One critical component of most microwave tomographic imaging techniques, and for that matter most inverse problems, is the Jacobian matrix [33]. While not called this exact term in many microwave formulations, the net result is similar for all techniques. In essence, the terms of the Jacobian are the change in the field registered by the *j*th receiver due to a small change in the permittivity at a single node in the parameter reconstruction mesh while illuminated by the *i*th transmitter [34]. These gradient terms provide direction to the iteration process during Newton-like reconstruction processes. The overall size of the matrix is $\left(N_{t}\times N_{r}\right)\times N_{p}$, where $N_{t}$, $N_{r}$, and $N_{p}$ are the number of transmitters, number of receivers per each transmitter, and the number of parameter values being reconstructed, respectively. In this sense, the matrix can be quite large and its formation can impose a significant impact on the overall computation times.

Cui et al. [35] introduced a Green’s function with an integral formulation technique to effectively form the Jacobian matrix which is subsequently incorporated into a distorted Born approximation inverse solution. The technique has been shown to work well with low contrast and small scatter problems when the starting property distribution estimate is that of the surrounding background. This approach is subsequently utilized by Shea et al. [18] and further by Karadima et al. [23]. Zakaria et al. [36] used the finite element contrast source inversion method for microwave imaging and reported in detail on the gradients of the cost function. Van den Berg and Kleinman [37] also devised a contrast source inversion method to solve the inverse problem. While this did not explicitly compute a Jacobian matrix, it does compute the gradients of the cost function which are in the form of Frechet derivatives, called the Polak–Ribiere conjugate gradient directions. This approach is further used by Gilmore et al. [25] which expands upon this concept by adding a multiplicative regularization. The contrast source inversion technique is also used by Rubaek et al. [38] which expands the concept to a log-transformed version, similarly to that used in Meaney et al. [39]. The conversion of the complex Jacobian terms to the log-transformed counterpart involves only minor algebraic operations. Joachimowicz et al. utilizes an iterative variation scheme derived from a classical Newton procedure [40]. The solution was derived for the 2D TM situation and the computation of the Jacobian matrix is a Green’s function formulation applied to a Method of Moments technique for the forward solution. Efforts by Souvorov et al. [34] utilized an iterative Fourier inversion method where the forward solution was based on integral equations. The Jacobian matrix formulation was quite similar to that of Cui et al. [35]. Later efforts by Bindu and Semenov [41] utilized the distorted Born approximation with an FDTD forward solution and similar Jacobian formulation. In none of these cases did the groups actually call their matrices the Jacobian matrix.

More recently, the discrete dipole approximation (DDA) has been introduced as a means of dramatically reducing the forward solution computation time and the associated memory requirements [42]. This has the potential to reduce the forward solution time of a 2D imaging problem by an order of magnitude or more. Interestingly, the uniform grid configuration informed our understanding of the Jacobian formulation to the point where its construction is now reduced to a low number of vector–vector multiplications. In a more rigorous analysis, this new derivation originates from the nodal adjoint technique introduced by Fang et al. [43]. The net result is that each element of the matrix is created by the multiplication of two complex numbers and a pre-computed scalar term. The adjoint method is an exact method for computing the Jacobian matrix and relies largely on the principle of reciprocity. It is widely used in multiple imaging modalities including optical coherence tomography and electrical impedance tomography in addition to microwave imaging [44,45,46,47,48]. This approach is quite general for the 2D problem and can be readily extrapolated to the 3D situation.

The paper focuses on the calculation of the Jacobian matrix as an expansion of the nodal adjoint technique with respect to its accuracy and computational cost. A comparison between Jacobian matrices formed via the finite element-type (FE) and the discrete dipole approximation-based (DDA) tomographic image reconstruction algorithms is performed. Our goal is to retain the efficiency embedded in the DDA forward solver and does not allow the Jacobian matrix formation to undermine the capabilities of the fast forward solver. In this study, we present an approach which utilizes the DDA forward solution configuration with the nodal adjoint formulation of the Jacobian matrix allowing us to preserve both performance and accuracy. The accuracy of the Jacobian matrix is investigated with respect to our FE-based version. Additionally, experimental data acquired using the microwave breast imaging system at Chalmers University of Technology is used to image a tissue-like phantom to validate our algorithm.

The following sections describe the mathematical underpinnings of the method and show examples of Jacobian matrix construction in comparison to an existing finite element-based formulation together with reconstructed images employing the described technique. The paper is organized in the following format: Section 2 describes the nodal adjoint method used for calculation of the Jacobian matrices of the FE and the DDA forward solvers together with investigations of their computational costs from a mathematical point of view; Section 3 shows simulated numerical results from the Jacobian matrices of the FE and DDA solvers, the computation times for these methods as well an image reconstruction validation by phantom experiment; Section 4 discusses the results and concludes the study.

## 2. Methods

Beginning with the finite element representation of the dual mesh-based nodal adjoint method derivation of the Jacobian matrix [43], Figure 1 shows the overlapping forward solution mesh (red) and the parameter mesh (blue). For the 2D case, all of the forward solution and parameter mesh elements are triangles. This discussion is readily extended to the 3D case by replacing the triangles with tetrahedrons and expanding the forward solution from this scalar problem to the necessary 3D vector forward distributions. In previous implementations, the size of the forward solution mesh elements was noticeably smaller than that for the parameter mesh because the former’s size is dictated by the sampling requirements to achieve an accurate forward solution [49].

In this case, a single term of the Jacobian associated with the *s*th source antenna, the *r*th receive antenna and the τth parameter mesh node can be written as follows:(1)$$J\left(\left(s,r\right),\tau \right)=-\sum_{n\in \Omega_{\tau }}\sum_{e\in \Omega_{n}}\frac{1}{j\omega \mu_{0}\left|J_{r}\right|}\frac{A_{e}}{M}\phi_{\tau }\left({\vec{p}}_{n}\right)E_{s}\left({\vec{p}}_{n}\right)E_{r}\left({\vec{p}}_{n}\right)$$
where $E_{s}\left({\vec{p}}_{n}\right)$ and $E_{r}\left({\vec{p}}_{n}\right)$ are the electric field values due to sources from both antennas *s* and *r* at the nodal location ${\vec{p}}_{n}$, and $\phi_{\tau }\left({\vec{p}}_{n}\right)$ is the parameter mesh basis function (In this situation, we utilize linear basis functions over each triangular element where its value is 1 at the designated node and transitions linearly to zero at the other two nodes and along the edges between these two nodes) due to node τ in the parameter mesh and evaluated at node *n* in the forward solution mesh. $\Omega_{n}$ is the region composed of forward solution elements immediately associated with node *n* (i.e., for the region where the forward solution basis function for node *n* is non-zero) (Figure 2). *e* is one of the forward solution elements within $\Omega_{n}$, $A_{e}$ is the area of element *e* and *M* is the number of vertices in a triangle (3). ω is the angular frequency, $\mu_{0}$ is the free space magnetic permeability, $J_{r}$ is the amplitude of a signal transmitted by the receive antenna *r*, and *j* is the imaginary unit. $\sum_{e\in \Omega_{n}}\left(\;\;\;\right)$ is essentially a numerical integration of the associated terms over element *e*. $\Omega_{\tau }$ is the region composed of parameter elements immediately associated with node τ (i.e., for the region where the parameter basis function for node τ is non-zero) (Figure 3). It should be noted that forward solution node *n* must lie within $\Omega_{\tau }$.

As can be seen from Figure 2, $\phi_{\tau }$ is non-zero for all seven forward solution nodes in $\Omega_{n}$. Therefore, these summations in Equation (Equation 1) need to be performed explicitly. However, an interesting observation is made if both the forward solution and parameter meshes overlap exactly (Figure 4).

In this case, there are six parameter mesh elements surrounding node τ and only seven forward solution mesh nodes within $\Omega_{\tau }$. In fact, for the integration (summations) which is performed over just the six τ-associated elements, the terms are only non-zero for node τ and zero for all the surrounding ones (Again, this is true for the linear basis functions used in this derivation). In this case, the inner summation can be reduced:(2)$$\frac{1}{j\omega \mu_{0}\left|J_{r}\right|}\frac{1}{M}\sum_{e\in \Omega_{n}}A_{e}\phi_{\tau }\left({\vec{p}}_{n}\right)E_{s}\left({\vec{p}}_{n}\right)E_{r}\left({\vec{p}}_{n}\right)=\frac{1}{j\omega \mu_{0}\left|J_{r}\right|}E_{s}\left({\vec{p}}_{\tau }\right)E_{r}\left({\vec{p}}_{\tau }\right)\sum_{e\in \Omega_{n}}\frac{A_{e}}{M}$$

For the situation where the mesh is effectively uniform, the new summation becomes a constant, where *A* is the sum of the area in $\Omega_{\tau }$. This ultimately reduces the Jacobian term to
(3)$$J\left(\left(s,r\right),\tau \right)=\frac{-1}{j\omega \mu_{0}\left|J_{r}\right|}\frac{A}{M}E_{s}\left({\vec{p}}_{n}\right)E_{r}\left({\vec{p}}_{n}\right)$$

This means that each term in the Jacobian is reduced to a simple multiplication of the two forward solutions due to sources from antennas *s* and *r* at node τ and also multiplied by a constant, $\frac{1}{j\omega \mu_{0}\left|J_{r}\right|}\frac{A}{M}$. It should be noted that for an entire row of the Jacobian matrix (i.e., where *s* and *r* do not change), each element of the matrix is a similar multiplication at different nodes of the mesh. This implies that an entire row of the Jacobian can be computed as a vector–vector multiplication of the two field distributions times a constant:(4)$$\bar{J}\left(s,r\right)=\frac{-1}{j\omega \mu_{0}\left|J_{r}\right|}\frac{A}{M}{\vec{E}}_{s}{\vec{E}}_{r}$$

Thus, computation of the entire Jacobian matrix can be reduced to $n_{s}\times n_{r}$ vector–vector ($n_{p}$ long) multiplications where $n_{s}$ and $n_{r}$ are the number of transmitting antennas and the number of receiving antennas per each transmitter, and $n_{p}$ is the number of nodes in the parameter mesh. In comparison, the computation for a single element of the Jacobian matrix using the nodal adjoint method is the product of two $n_{f}$ long vectors and a corresponding diagonal matrix, where $n_{f}$ is the number of nodes in the forward solution mesh. This amounts to $2\times n_{f}$ computations for a single element and $2\times n_{f}\times n_{p}$ computations for a complete row. Therefore, the new approach has a factor of $2\times n_{f}$ fewer computations—corresponding to several orders of magnitude faster construction. This is a general relationship and it can be easily demonstrated that it holds for other geometric configurations such as rectangular grids used in finite difference time domain and the discrete dipole approximation.

## 3. Results

### 3.1. Comparison of Jacobian Matrix Row Distributions with That of Known Reconstruction Algorithm

Figure 5 shows a schematic diagram of a typical 2D imaging system arrangement. In this case, the monopole antennas are positioned on a 15.2 cm diameter circle, with antenna #1 being the lowest centered one and the subsequent ones ordered in a clockwise configuration. Each antenna broadcasts a signal for which the complementary 15 antennas act as receivers. This process is repeated for each antenna transmitting individually to produce a total of 240 measurements (16 transmitters × 15 receivers per transmitter).

Figure 6a shows the imaging zone for the discrete dipole approximation (DDA) method where the circular imaging zone is approximated by a subset of the uniform grid used for computing the forward solution. In this case, the complete grid is comprised of 64 × 64 squares for a total forward solution zone of 25 × 25 cm${}^{2}$ and each single cell is a 3.9 mm square.

The imaging zone is a step-wise circle consisting of 1012 squares and 1085 vertices (i.e., the number of unknowns) with an effective diameter of 14.0 cm. Figure 6b,c show the corresponding finite element (FE) meshes used in our FE-based reconstruction algorithm. As with the DDA approach, the antennas are configured on a circle concentric with the centre of the imaging zone. The meshes for the dual mesh approach (one for the forward solution and one for the property reconstruction) are both inside the antenna array with diameters of 14.5 cm. It should be noted that for the forward solution, the region outside of the meshes is represented using the boundary element method comprised of the homogeneous medium and having outer boundaries extending to infinity [50]. The fine mesh (forward solution) has 3903 nodes and 7804 triangular elements while the coarse one (reconstruction) has 559 nodes and 1044 elements. The average node spacing for these two meshes is 2.25 mm and 6.08 mm, respectively.

Figure 7 shows the plots of several rows of the Jacobian matrix using the DDA algorithm on the 1085 node grid corresponding to different transmit and receive antenna pairs at 1300 MHz and homogeneous bath properties of $\epsilon_{rb}$ = 22 and $\sigma_{b}$ = 1 (S/m). These calculations all assume that this is the first iteration of a reconstruction problem where the starting distribution of the imaging zone is homogeneous utilizing the known values of the bath.

In this case, the complex values are transformed into their effective log magnitude and phase values via their transformation relationships described in Meaney et al. [39] for viewing their distributions. The different cases include: (a) transmitting at antenna #1 and receiving at antenna #8, (b) transmitting at antenna #5 and receiving at antenna #13, and (c) transmitting at antenna #6 and receiving at antenna #12, respectively. For the magnitude plots, there is a cigar-like yellow and red zone extending straight between each of the associated antennas and decreasing rapidly and oscillating about zero in the orthogonal direction to both sides with respect to the main axis of the band. Given that each matrix term is the change in the field measured at the receiver with respect to a change in the properties at a pixel for a signal broadcast at a transmitter, these distributions effectively represent sensitivity maps for the given transmit/receive pair antennas. It is important to note that these distributions are reciprocal—i.e., the distribution for transmitting from antenna *A* to *B* is the same as that for transmitting from *B* to *A*. Reciprocity is a key feature in exploiting the adjoint technique. The corresponding phase plots show a similar but wider main band extending between the associated antennas. Extending outwards in the orthogonal direction from this main band, the phase values increase from negative values and then oscillate about zero outwards while diminishing in amplitude.

Figure 8 shows the corresponding distributions for the FE-based algorithm plotted on the associated 559 node mesh. The overall effective magnitude and phase distributions are quite similar. The overall scaling is slightly different because the individual values of the Jacobian matrix terms are with respect to the effective grid area immediately surrounding each pixel in the imaging zone. For the FE-based arrangement, the area is 96.0 mm${}^{2}$ while that for our uniform grid/DDA configuration, it is 60.8 mm${}^{2}$.

Other than this scaling factor, the results are essentially identical.

Figure 9 shows the maps of each row of a FE-based Jacobian matrix divided by the associated DDA rows (both effective log magnitudes and phases).

In these cases, the FE-based distributions were first mapped to the DDA-based grid using a linear interpolation and divided pixel-by-pixel by the DDA distributions. The results are nearly uniform distributions across the domain except for large and isolated perturbations along arcs outside of and following the trajectory of the main bands within both the associated magnitude and phase distributions. The two distributions differ by a scaling factor which is related to the fact that the local area in the imaging zone is different because the number of nodes is different for each implementation. This area term has an immediate impact as can be seen in Equation (Equation 1). These jumps are caused by instances where the values in the DDA-based distributions are near zero (along the loci of points where the distributions cross zero) while the mapped version of the FE-based distribution is slightly altered spatially (still crossing zero), resulting in these disruptions. In spite of these aberrations, which are minor, the overall uniform distributions confirm the excellent match between the Jacobian calculations for the two methods.

### 3.2. Computation Time

For the computation time, we only considered the recurring steps at each iteration. For instance, the computation time for the weighting factors that only needed to be calculated once and stored in memory was not counted. For the FE-based approach, the code was written in FORTRAN77 and was compiled with gfortran (v. 7.5.0) and running on a single 4-core Xeon E3-1270 3.60 GHz processor with hyperthreading enabled and 32 GBytes of DDR4 2133 MHz RAM running the Ubuntu (18.04 LTS) operating system. For the DDA-based algorithm, the software code was written in MATLAB (v. R2019b MathWorks, Inc., Natick, MA, USA) and ran on the same machine. The average time (averaged over 20 iterations) for computing the FE-based Jacobian matrix was 1.08 s. The corresponding time for the DDA-based code was 0.0055 s. It is important to note that the sizes of these two matrices are different. While the number of rows for each is the same (16 transmit antennas × 15 receive antennas per each transmitter), the number of nodes in the field of view (i.e., the number of columns of the matrix) is different. For the FE-based scheme, the number of columns was 559 while that for the DDA-base method was 1085. Even with this considerable difference, the new DDA-based approach was substantially faster.

### 3.3. Experimental Validation

The 2D DDA forward solver and the corresponding Jacobian matrix are embedded in the log-magnitude and phase format reconstruction algorithm. The fundamental aspects of the reconstruction algorithm and the 2D DDA were previously investigated by the authors [27,42,51,52] and are employed for this study.

The microwave breast imaging system at Chalmers University of Technology [53] is comprised of an imaging tank filled with a coupling medium, i.e., mixture of glycerin and water (80:20 ratio) and a surrounding circular antenna array comprised of 16 monopole antennas. The lossy mixture is a tissue mimicking medium and resembles the breast tissue for purposes of imaging. The 4 cm diameter cylindrical phantom filled with 84:16 glycerin–water mixture is positioned in front of antennas #6 and #7, shown in Figure 10. The scattering parameters of the system were measured via the Vector Network Analyzer (Rohde & Schwarz ZNBT8) connected to the antenna array, and are provided as the measurement data to the reconstruction algorithm in form of two sets of 240 (16 transmitter × 15 receiver per transmitter) measurements (magnitude and phase parts of the electric fields).

At 1500 MHz, the dielectric properties for the 80:20 water–glycerin mixture used in the imaging tank was measured such that the relative permittivity $\epsilon_{r,bath}=20.9$ and conductivity $\sigma_{bath}=1.35$ (S/m). The corresponding values for the 84:16 water–glycerin mixture used in the cylindrical phantom (object) are $\epsilon_{r,obj}=16.9$, $\sigma_{obj}=1.15$ (S/m).

Figure 11 shows the reconstructed relative permittivity and conductivity images as a function of iteration number. The circular object is visible in both permittivity and conductivity images in all iterations at an off-centered position (a circle is drawn to show the exact location of the object). The sequence of images illustrates how the algorithm recovers the object in the first few iterations and then gradually refines it. We observe that the reconstructed permittivity images are more accurately recovered compared to those of the conductivity, especially the conductivity images have more artefacts surrounding the phantom position while the permittivity images are not as affected by the artefacts and are smoother. This observation is in line with previous investigations assessing image quality [24,54].

To confirm the convergence of the reconstruction algorithm, we compare the field calculations with the actual measured data. The field projections refer to the calibrated data as the measured data for the homogeneous imaging tank (without the phantom) subtracted from the inhomogeneous measured data (containing the phantom). Figure 12 shows the computed magnitude and phase projections due to a single transmitter at selected iterations compared with that of the actual measured data. The projections are of similar shapes as the measured data at all iterations and monotonically converge towards them.

Figure 13 shows the error of the optimization process for image reconstruction. The relative error (normalized to the first iteration) decreases monotonically with increasing iterations. In this case, the error decreased from 1 to 0.135 in 18 iterations. The stopping criterion is met when the decrease in error from one iteration to the next is less than 1 × 10${}^{-3}$.

## 4. Discussion and Conclusions

For microwave tomographic type imaging algorithms, construction of the Jacobian matrix contributes significantly to the overall computation time. It is computed at each reconstruction iteration, so its time impact is important. Previous imaging techniques have employed the adjoint method which exploits reciprocity to dramatically reduce the computation time. For those implementations, the reconstruction algorithm was finite element based, such that computing the Jacobian meant performing numerous integrations over the various triangular elements and sub-elements within the imaging zone just to calculate a single value within the matrix. The net consequence was that the time was still substantial. In this formulation, we exploit the observation that the distributions of rows of the Jacobian produce effective sensitivity maps with respect to the pair of transmitting and receiving antennas. Inspection of the adjoint-based algorithms used to produce the matrix shows that when the parameter mesh overlaps exactly with the forward solution mesh, the net effect is that the sensitivity distributions are simplified to simple multiplications of the field vectors due to sources at both the transmit and receive antennas. This allows each row of the matrix to be computed as a simple vector–vector product of the two forward solutions which is a dramatically smaller computation task than before.

It should be noted that this approach could be adapted to situations where the forward solution mesh is not identical to the parameter reconstruction mesh. In those situations, the rows of the Jacobian could be computed using this approach as if with the assumption that the two meshes are the same. Subsequently, this distribution could then be mapped from the one mesh to the other utilizing a simple matrix-vector multiplication. While this would involve extra computational steps compared to this preferred implementation, it would most likely be more efficient than existing methods for computing the Jacobian matrix.

For many algorithms, the computation time for constructing the Jacobian matrix usually consumes the second most amount of time behind that for computing the forward solutions. However, with the advent of the discrete dipole approximation approaches, the forward solution times are reduced dramatically—to the point where the times for constructing the Jacobian matrix may become comparable. In keeping with reducing the overall time, it is imperative that this task not be ignored. Our new treatment allows the Jacobian matrix computation to be substantially reduced and has a direct impact on the overall image reconstruction time without sacrificing accuracy.

The reconstructed images from the experimental study confirm that the simple and efficient calculation of the Jacobian matrices from our new approach and the DDA forward solutions produce high quality images in a highly efficient manner.

## Figures and Tables

**Figure 1 sensors-21-00729-f001:**
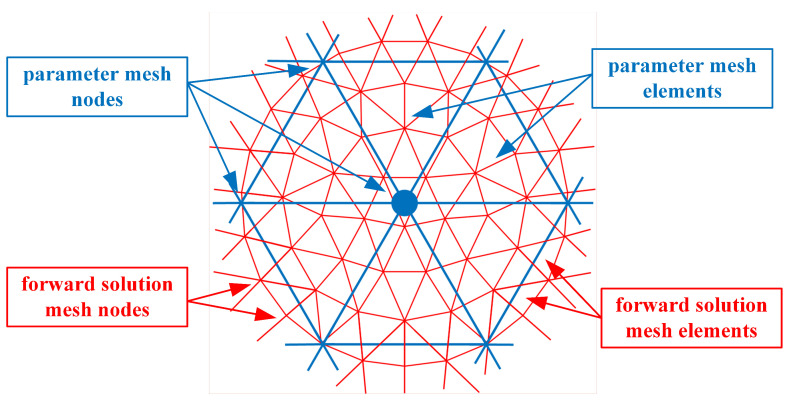
Overlapping forward solution and parameter meshes corresponding to the nodes and elements.

**Figure 2 sensors-21-00729-f002:**
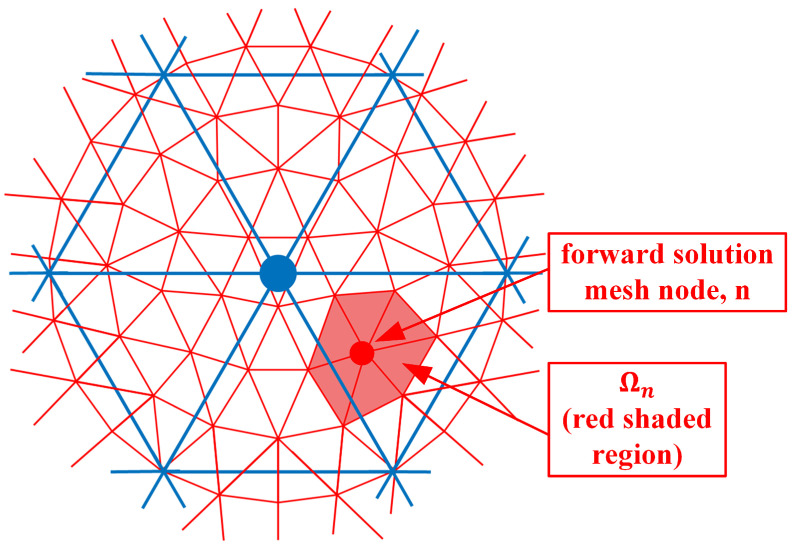
Forward solution mesh nodes; example of node *n* and corresponding region $\Omega_{n}$ is sketched.

**Figure 3 sensors-21-00729-f003:**
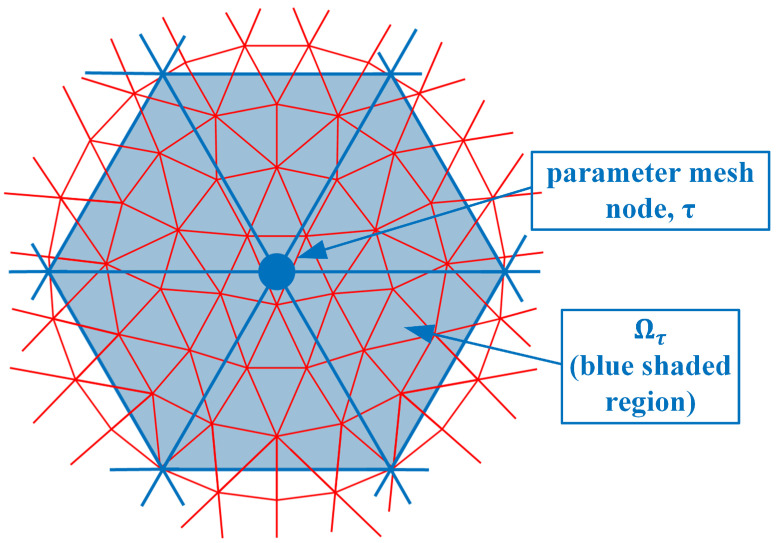
Parameter mesh nodes; example of node τ and corresponding region $\Omega_{\tau }$ is sketched.

**Figure 4 sensors-21-00729-f004:**
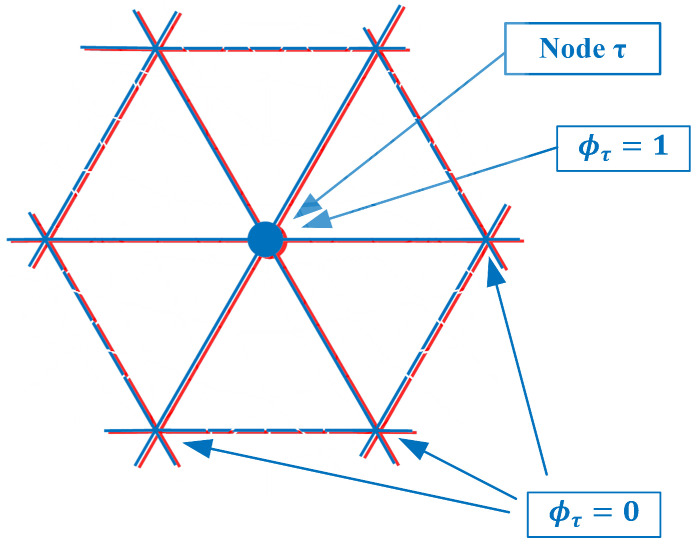
Forward solution mesh and parameter mesh overlap exactly.

**Figure 5 sensors-21-00729-f005:**
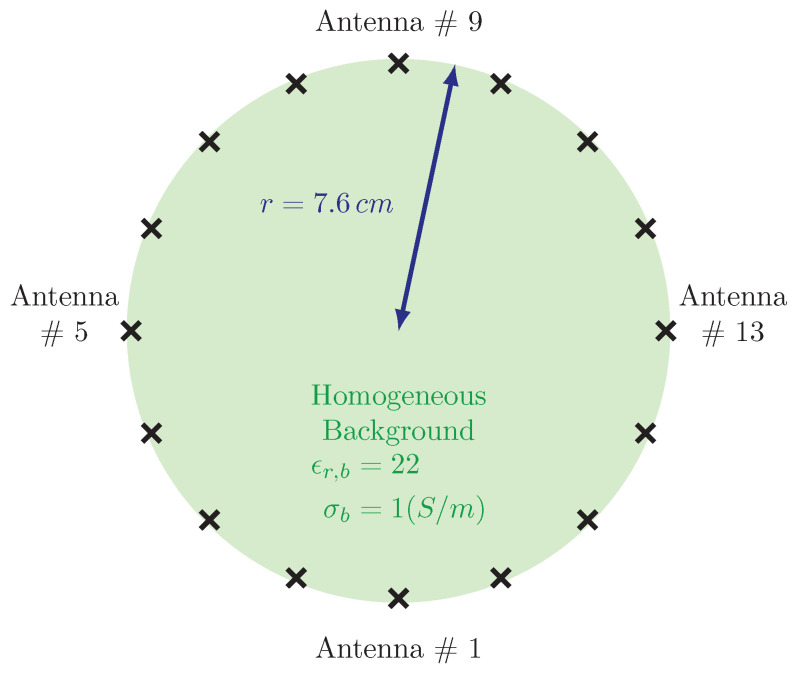
Schematic representation of imaging domain together with the antenna array.

**Figure 6 sensors-21-00729-f006:**

Representation of imagining domain with respect to the meshes used for forward solutions (**a**,**b**) and property reconstruction (**a**,**c**).

**Figure 7 sensors-21-00729-f007:**
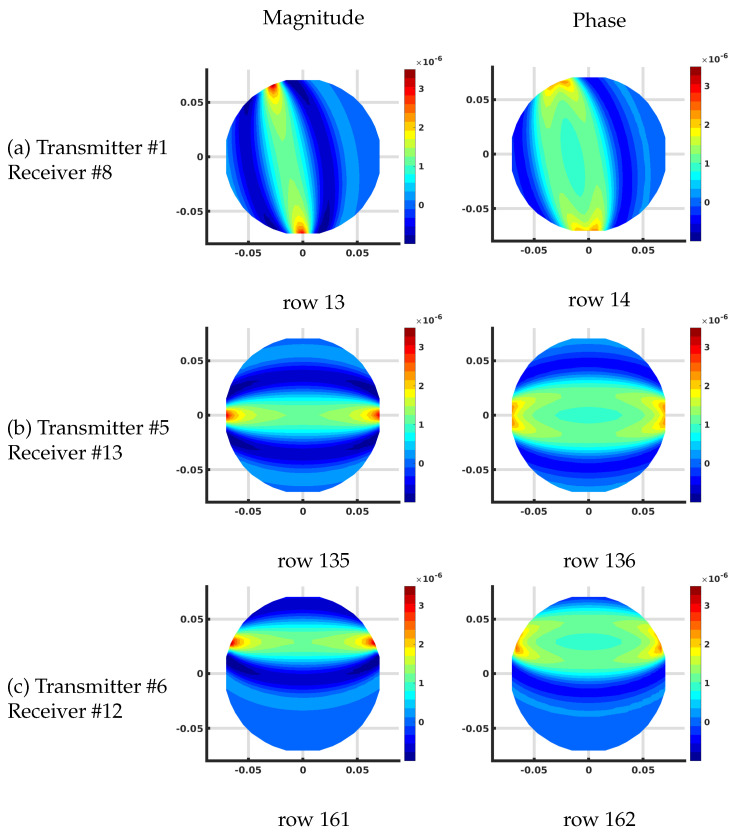
Plots of the effective log magnitude (**left**) and phase (**right**) distributions for various rows of Jacobian matrix (m${}^{2}$), corresponding to multiple transmitter/receiver pairs, for the discrete dipole approximation (DDA)-based algorithm. The distributions are presented on the DDA grid. Note that the plotting algorithm smoothed the imaging zone shape with triangles around the sharp edges of the reconstruction grid.

**Figure 8 sensors-21-00729-f008:**
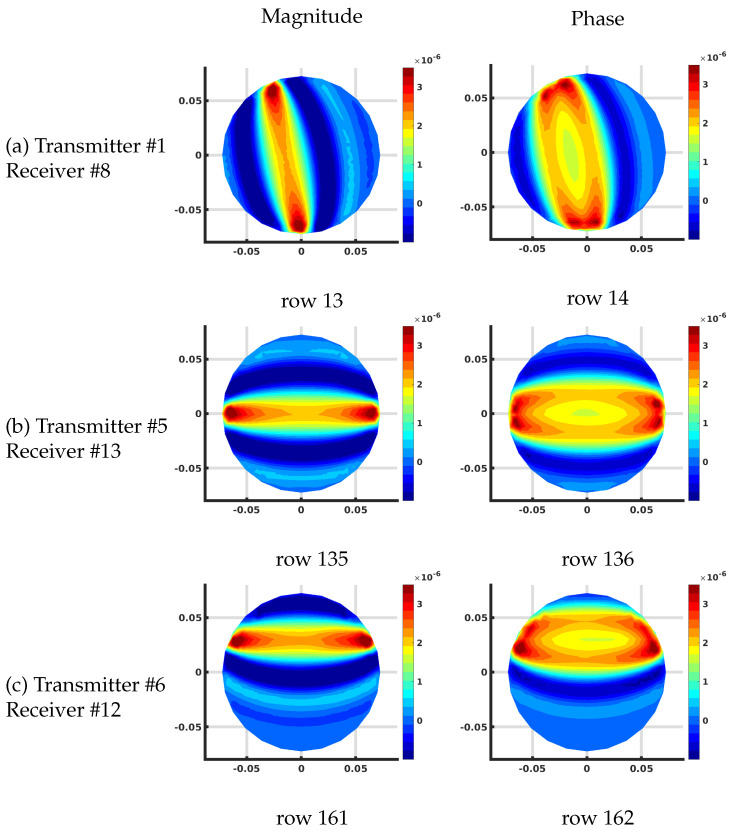
Plots of the effective log magnitude (**left**) and phase (**right**) distributions for various rows of Jacobian matrix (m${}^{2}$), corresponding to multiple transmitter/receiver pairs, for the finite element (FE)-based algorithm. The distributions are presented on the coarse FE-mesh.

**Figure 9 sensors-21-00729-f009:**
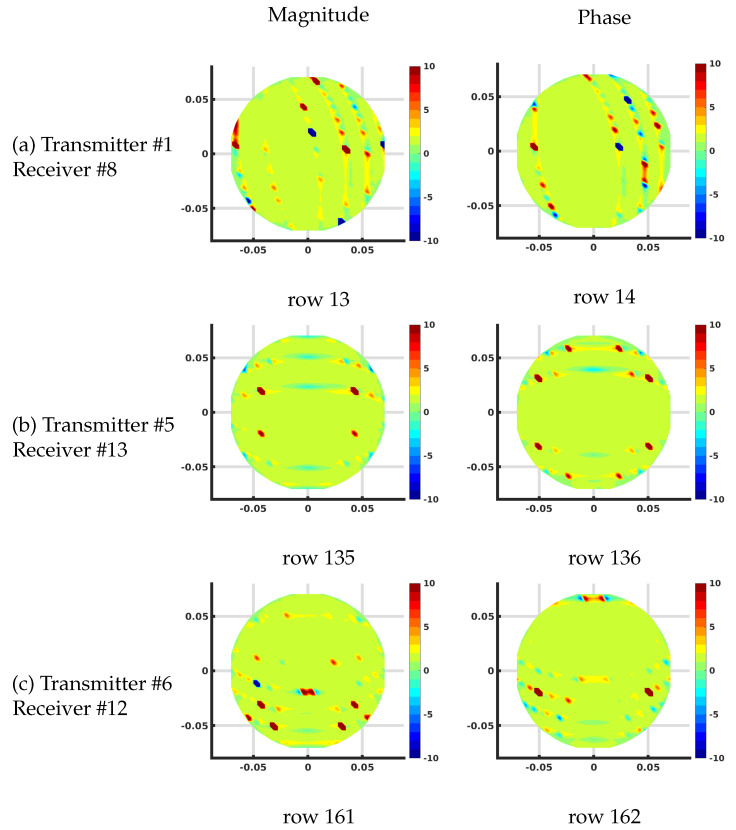
Plots of the effective log magnitude (**left**) and phase (**right**) distributions for ratios of Jacobian matrices calculated with DDA and FE-based schemes. The distributions are presented on the DDA grid.

**Figure 10 sensors-21-00729-f010:**
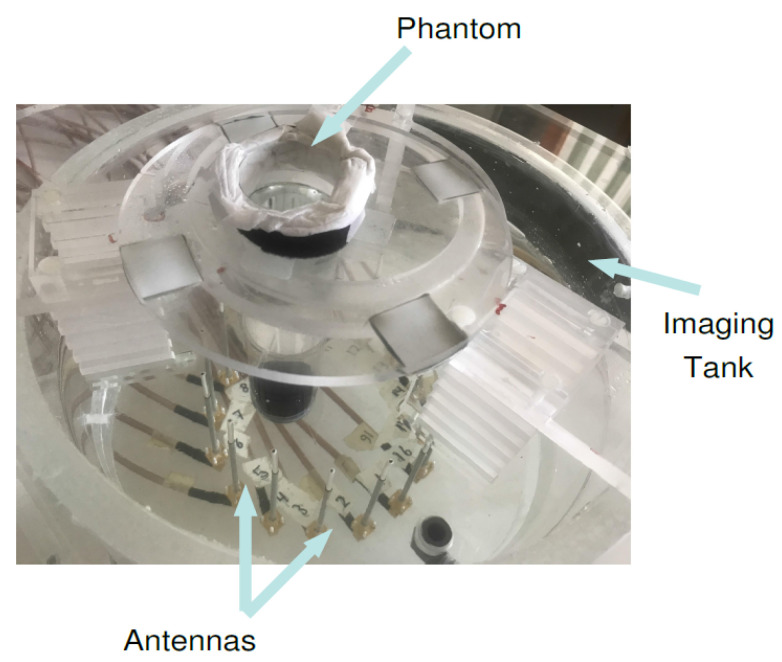
The photograph of the measurement setup for the case that a cylindrical inclusion is positioned close to antennas #6 and #7.

**Figure 11 sensors-21-00729-f011:**
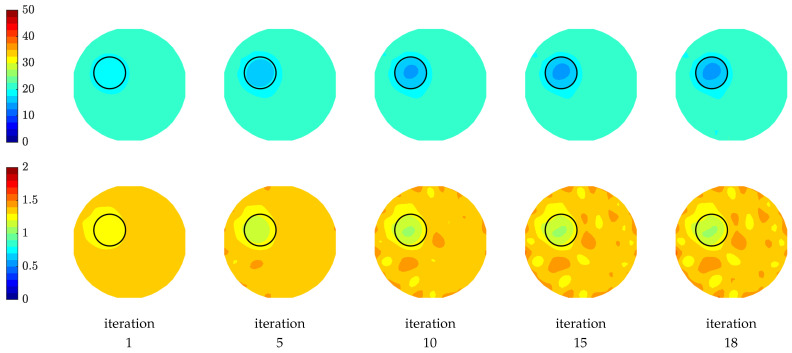
Images of reconstructed relative permittivity (**top row**) and conductivity (S/m) (**bottom row**) at 1500 MHz as a function of iteration number.

**Figure 12 sensors-21-00729-f012:**
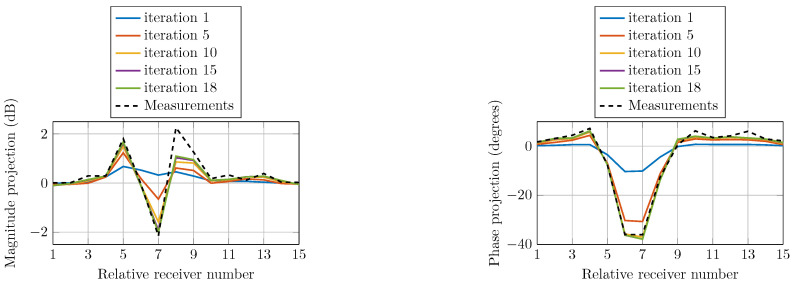
Comparison of calculated magnitude and phase projections of the signals transmitted from Antenna 1 for multiple iterations as a function of receiver number. The actual measurement projection is also shown.

**Figure 13 sensors-21-00729-f013:**
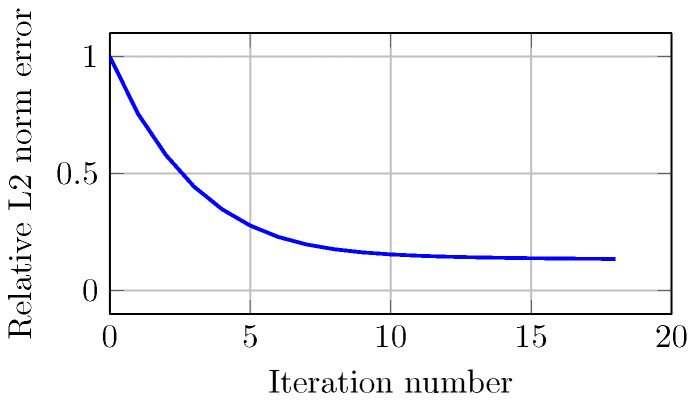
Computed relative L2 norm error of projections for the experimental study as a function of number of iterations.

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
