# Peer review of "Expansion of the Nodal-Adjoint Method for Simple and Efficient Computation of the 2D Tomographic Imaging Jacobian Matrix"

_sensors, 2021, doi:10.3390/s21030729_

Round 1

Reviewer 1 Report

The work is interesting and well presented. I would have some questions. 

  • Maybe I got the wrong reading, but the statement "In this way, the nodal adjoint method for computing the Jacobian matrix decreases to Order N computations for each row of the matrix" in the abstract gives me the idea of an N^2 complexity. Which would be the complexity of the alternative calculation?

  • In lines 104 (" This approach is quite general for the 2D problem and can be readily extrapolated to the 3D situation.") and  109 ("This discussion is readily extended to the 3D case by replacing the triangles with tetrahedrons"), the authors talk of the easy implementation on 3D. Do the authors have any comment on the fact that the field would be vectorial? Does the scheme remain simple enough? 

  • Line 114: "parameter mesh basis function". Maybe it is implicit, but it would be useful for the reader to have a definition of the basis function while following the explanation. (Is it a pyramidal basis function?)

  • Does in some way the increase of the number of nodes in the Jacobian calculation (due to the use of the same forward mesh) limit the gain of the proposed scheme? (If it is the forward mesh that is going towards the Jacobian mesh... there's no accuracy drawback?) 

Some comments regarding the plots: 

  • Which are the units of the phase? It's not clear why it is the same scale of magnitude. 
  • (Is magnitude dimensionless due to the normalization?) 
  • It's not clear for me why magnitude and phase (each column of the plots) are related to 2 different and consecutive rows (Is it due to the way data is saved?)
  • In lines 171-174, the authors refer to a), b), and c). No equivalence is found in the plots. 
  • Figure 9 refers to a ratio. I would expect a magnitude close to 1 while the phase tends to zero in the case of high similarity. Maybe I missed something in the explanation. 
  • In Figure 9, the plots called "row 13" and "row 14" (or magnitude and phase) look like the same image. 
  • It would be useful to define a metric that better explains how good figure 9 is. 

Reviewer 2 Report

This paper presents a method/scheme of calculating Jacobian matrix for solution of microwave inverse imaging problem. The idea presented provides a simplified (but trivial) way of estimating the Jacobian using the Discrete Dipole Approximation for forward solution and similar meshes for the parameters as well as forward solution.

The reviewer has noted following concerns:

  • The work presented, though interesting, is not a significant contribution in the field
  • The background and introduction should extend to other significant research in microwave imaging
  • The key contribution of this work is not clear. Comments and comparison with other relevant work (in the area of computational enhancements) should be presented
  • The proposed approach needs to be tested and validated in an appropriate application (e.g. microwave breast imaging or brain imaging)  

Round 2

Reviewer 1 Report

I want to thank the authors for their precise and extensive answers. 

The work is interesting, and the new addition in the results section improves soundness. 

I still have a comment relative to the log-phase plots. I understand that the units of the Jacobian are m2. Still, the phase (by definition) should have an angular unit. Probably I misunderstand something. 

In any case, my comment on magnitude 1 and phase 0 was related to the RATIO between Jacobians (Figure 9).  If both of the compared values are quite similar (as is expected to demonstrate), should not this ratio tend to 1? (or to a constant value if there is some extra factor in the middle, as the uniformity described by the authors.)

Should not the ratio be dimensionless? (as Figure 9 caption confirms)

(These comments are related to the author's answer: "They have units of m2 so 1 and 0 are not representative values for comparison in this case. ")

I apologize for insisting on this point. It is not so clear to me. 

Reviewer 2 Report

I believe the revised manuscript has improved with addition of a use-case of the proposed method, and other revisions have also improved the overall quality of the manuscript. 

Author Response

We thank the reviewer for the evaluation and feedback. There is no point to address in this revision concerning Reviewer 2.